# A new look at reflection seismic data from the Central Caledonian Transect across the Scandinavian Peninsula

Christopher Juhlin[1], Rodolphe Lescoutre[2], Bjarne Almqvist[1]

[1]Dept. of Earth Sciences, Uppsala University, Uppsala, Sweden
[2]Mantle8 sas, 6 rue de Chamechaude, 38360 Sassenage, France

*Correspondence to*: Christopher Juhlin (christopher.juhlin@geo.uu.se)

**Abstract.** This study revisits seismic reflection data from the central Scandinavian Caledonides, initially acquired during campaigns in the late 1980s and early 1990s. Previous analyses faced challenges in merging and imaging due to varying trace spacing and data gaps, particularly in the western parts. To address these limitations, we spatially resampled the data to a consistent trace spacing, carefully merged segments, and migrated the entire merged section. This approach resulted in a revised seismic profile, with notable changes in the western section where the image reveals key differences compared to earlier interpretations. The updated profile indicates near-continuous reflections across merged segments, resolving issues of abrupt breaks present in some prior publications. Enhanced imaging in the western section unveils new structural details, including collapsed diffractions and shorter reflective segments, offset from one another. These reflecting segments in the Skardöra antiform are interpreted as representing dolerite sills that were once continuous over a larger area, but have been offset by normal faulting. This reinterpretation suggests a significantly thinner Upper Allochthon in the west than in previous interpretations. These results emphasize the importance of careful data integration and migration for seismic interpretation, shedding new light on the structural complexity of the western Scandinavian Caledonides. The study contributes to refining geological models and advancing understanding of the region's tectonic history.

## 1 Introduction

A major effort to acquire reflection seismic data across the central Scandinavian Caledonides was carried out in the late 1980s and early 1990s (e.g. Hurich et al., 1989; Palm et al., 1991; Juhojuntti et al., 2001). Results from these campaigns have been presented and interpreted earlier, in the form of line drawings or seismic sections, either in part or as a continuous profile. Lescoutre et al. (2022a) provides the most recent presentation of the entire merged data set, spanning a distance of over 200 km, from western Trondelag in Norway to east of the Caledonian Front in Sweden. In merging the data from the different campaigns one problem that has not been properly addressed is the lack of a comprehensive migrated image in the western part of the transect. This has been hampered by differences in trace spacing between the various campaigns and not handling the overlap and gaps in an optimal manner. In this short communication we present a new version of the data set in which we have spatially resampled all data to a fixed trace spacing, merged the various campaigns more carefully and then

migrated the entire seismic profile at one time. This results in a somewhat different image in the western part of the profile where some important details differ from previous presentations, while the eastern part of the seismic image is essentially the same as that presented in Juhojuntti et al. (2001) and Lescoutre et al. (2022a). In this paper we first review the available data, present the processing strategy and results, and finally discuss the implications for structural interpretation in the western part of the profile. Our new interpretation is aided by incorporating results from the two c. 2.3-2.5 km deep COSC boreholes that were drilled in the Swedish Caledonides in recent years (Lorenz et al., 2015; Lorenz et al., 2022). Even though the profile presented here does not directly pass over the boreholes we can make use of important observations from these boreholes. In particular, the strong reflections from the Precambrian basement observed on a high resolution seismic profile (Juhlin et al., 2016) passing over the COSC-2 borehole are generated by dolerites that have intruded into highly homogenous volcanic rocks (Lorenz et al., 2022; Lescoutre et al., 2022a).

## 2 Geological Setting

The structural setting of the central Scandinavian Caledonides has been studied for well over a century (Törnebohm, 1888; Gee and Stephens, 2020, and references therein). These studies have contributed to major advances in understanding of fold-and-thrust belts and more specifically to the understanding on how they were formed. Closure of the Iapetus Ocean during the Silurian period, leading to a full continent-continent collision between the paleocontinents Baltica and Laurentia in the early Devonian (~400 Ma), was the initial phase of their formation. Paleogeographic reconstructions (Torsvik and Cocks, 2017) suggest that subduction was mainly westward to WNW at ~425-420 Ma and switching to W to WSW at around 410-400 Ma. During the closure stage, sediments and sedimentary rocks from the Baltica passive margin were subducted and partially returned and thrust onto Baltica (Arnbom, 1980; Gee et al., 2008; Majka et al., 2014). A stack of thrusted sheets developed, where some of the units were transported more than 400 km onto Baltica (Gee and Stephens, 2020). This stack of allochthons is typically divided into different units depending on their position and origin in the stack, and consist of the Uppermost, Upper, Middle and Lower allochthons, which overly the parautochthonous/autochthonous Precambrian basement (Table 1).

Starting at the stratigraphically uppermost unit, the Uppermost allochthon comprises rocks of Laurentian affinity. The Upper allochthon, including the Köli Nappe complex, consists of remnants of oceanic crust in the form of partly metamorphosed gabbros and metamorphosed sedimentary rocks (i.e., phyllites). The Uppermost and Upper allochthons experienced greenschist grade metamorphism.

Continuing down the tectonostratigraphy, the Middle allochthon (including the Seve nappe complex) forms the central part of the stack. These rocks experienced the highest-grade metamorphic conditions in the entire stack, with widespread evidence for granulite facies conditions in the middle part of the Seve nappe complex (Arnbom, 1980), and more recent evidence for ultrahigh pressure conditions as indicated by the presence of microdiamond inclusions in garnet (Majka et al., 2014; Klonowska et al., 2017). The Lower Seve Nappe complex of the Middle allochthon mainly consists of para- and

orthogneisses formed at amphibolite grade conditions. Jeanneret et al. (2022) recently used Ti-in-quartz geothermobarometry to constrain the peak metamorphic conditions of the Lower Seve to upper amphibolite/lowermost eclogite facies conditions, exceeding 1 GPa pressure and 600 °C. The lowermost parts of the Middle allochthon, consisting of the Särv and Offerdal

Nappes, have similar protolith as the Seve nappe complex, but experienced significantly lower grade metamorphic conditions. The sedimentary rocks that make up the Särv Nappe originate from the passive margin of Baltica and have been frequently intruded by mafic dolerites that were emplaced during the initial rifting and opening of the Iapetus Ocean ~600 Ma. Notably, metamorphosed equivalents of these dolerites exist also in the overlying Seve nappe complexes.

The Lower allochthon consists of a succession of sedimentary rock units that range in age from the Cryogenian (>700 Ma) to

70 Silurian (~420 Ma) periods. In the regions of the central Scandinavian Caledonides, this succession is also referred to as the Jämtlandian nappes. During orogeny, these sedimentary units formed parts of the foreland basin, with deformation taking place in the form of duplexing and local overthrusting of units within the Lower allochthon. The metamorphic grade of these units is generally greenschist facies. Tectonostratigraphically, the Alum shale makes up the lowermost part of the Lower allochthon, and marks the sole thrust or décollement of the nappe stack in the east. Although duplexing can make for some

75 complicated local tectonostratigraphic relationships, the Alum shale generally forms the boundary to the underlying autochtonous basement.

**Table 1: Tectonostratigraphy, Central Scandinavian Caledonides (Norway and Sweden) based on publications by Robinson and Roberts (2008), Gee and Stephens (2020), Saalman et al. (2021) and Jakob et al. (2022).**

| Tectonostratigraphic level | Sub-units | | |
|---|---|---|---|
| | *Norway (central/southern)* | *Sweden (Jämtland)* | *Original terrane* |
| Uppermost allochthon | Helgeland, Rödingsfjället, Fauske | Rödingsfjället | Laurentia affinity |
| Upper allochthon | Gula (Støren, Meråker Nappes) | Köli | Iapetus Ocean derived |
| Middle allochthon | Blåhø Nappe | Seve Nappe Complex | Baltica margin |
| | Sætra Nappe | Särv Nappe | |
| | Risberget Nappe | Tännäs Augen Gneiss Nappe | |
| Lower allochthon | Various, depending on location (Oyangen formation, Åmotsdal quartzite, Gjevilvatnet) | Jämtland Nappes | |
| Autochthon/para-autochthon | Baltica basement/gneiss | Precambrian granite, porphyry and gneisses | Crystalline basement |

The underlying Precambrian crystalline basement underneath the allochthonous units consists mainly of granitoids (granodiorite and quartzmonzonite) and volcanic porphyry (rhyolite, dacite, trachytes), which were recently dated by Andersson et al. (2022). The granites range in age from ~1690 to 1660 Ma, whereas the porphyries are significantly younger

with ages ranging from 1670-1650 Ma (Andersson et al., 2022). A detailed transect of the Lower Allochthon and crystalline
basement was recently obtained through the COSC-2 scientific drilling project (Lehnert et al., 2024; Lorenz et al., 2022).
The drilled section contains well-preserved felsic porphyries, whose age ranges from ~1660 to ~1650 Ma. The whole-rock
chemical composition of these porphyries are identical to porphyries originating from the Transcandinavian Igneous Belt
(TIB) in the nearby Fennoscandian Shield. These porphyries and related granitoids have been widely intruded by dolerite
dikes related to different intrusion generations, with the most abundant in central Sweden being the Central Scandinavian
Dolerite Group (CSDG) (Söderlund et al., 2006). Dolerite sheets identified in the COSC-2 borehole were dated using U-Pb
baddeleyite geochronology, with two sets of preliminary ages (Lescoutre et al., 2022b). The older age is ~1470 Ma,
predating the CSDG, whereas the second set of ages range from 1270-1260 Ma, overlapping with the CSDG. It is evident
that the crystalline basement has in places been involved in Caledonian deformation, such as the tectonic window that makes
up the Mullfjället antiform (Robinson et al., 2014). Brittle deformation observed in the basement shows in part Caledonian
ages, as demonstrated by K-Ar ages of fault gouges in basement just east of the present-day Caledonian foreland boundary
(Almqvist et al., 2023).

**Table 2: Overview of seismic acquisition. For source type: E – explosive, V – vibroseis.**

| Segment | A | B, C | D, H | E, G | F | I | J |
|---|---|---|---|---|---|---|---|
| Approximate length (km) | 45 | 19 | 15, 25 | 25, 18 | 6 | 50 | 51 |
| Source spacing (m) | 40 | 200 | c. 4800 | 400 | 200 | 50 | 50 |
| Source type | V | E | E | E | E | V | V |
| Receiver spacing (m) | 40 | 25 | 50 | 50 | 25 | 50 | 50 |
| Channels | 96 | 96 | 96 | 96 | 96 | 48 | 48 |
| Fold | 48 | 12 | 1 | 12 | 12 | 24 | 24 |
| Record length (s) | 10 | 20 | 20 | 20 | 20 | 10 | 10 |
| Sample interval (ms) | 4 | 2 | 2 | 2 | 2 | 4 | 4 |
| Year acquired | 1987 | 1987 | 1988 | 1988 | 1987 | 1990 | 1992 |

## 3 Data

Figure 1 shows the general geology of the study area and the locations of the profiles included in this work. We have
followed the labelling of Palm et al. (1991) for the different segments for the data acquired in 1987 and 1988. For the
extensions to the east we have labelled the 1990 extension as segment I and the 1992 extension as segment J. For the
segment west of Meråker, Norway (referred to as the Western Half in Roberts and Hurich (2018)), we have labelled this

segment A. Digital stacked data exist further west of segment A, but have not been included in this contribution. Acquisition parameters for the various segments are summarized in Table 2. As can be noted in Table 2 and in Figure 1, the greatest variability and overlap between segments can be found in the western part of the survey (segments A to F). A more detailed overview of this area is shown in Figure 2.

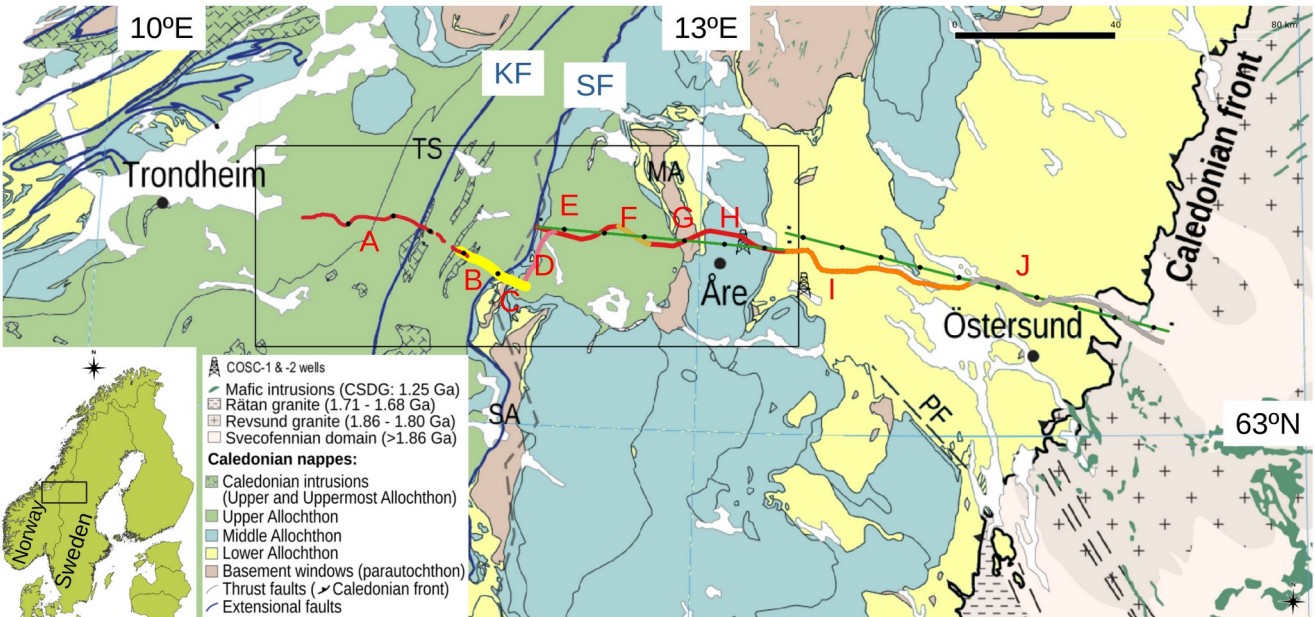

**Figure 1: Location map showing the different segments discussed in this paper. The geological map is based on Lescoutre et al. (2022a). Segments with similar acquisition parameters are coded with the same colour (except for I and J). Segments A, B and C were processed along crooked lines following the acquisition roads while segments E to J were projected onto two straight lines (coloured green). Every 400ᵗʰ CDP is marked by a black dot. MA: Mullfjället antiform; OW: Olden Window; PF: Persåsen fault; SA: Skardöra antiform; TS: Trøndelag synform; KF: Kopperå Fault; SF: Steinfjell Fault.**

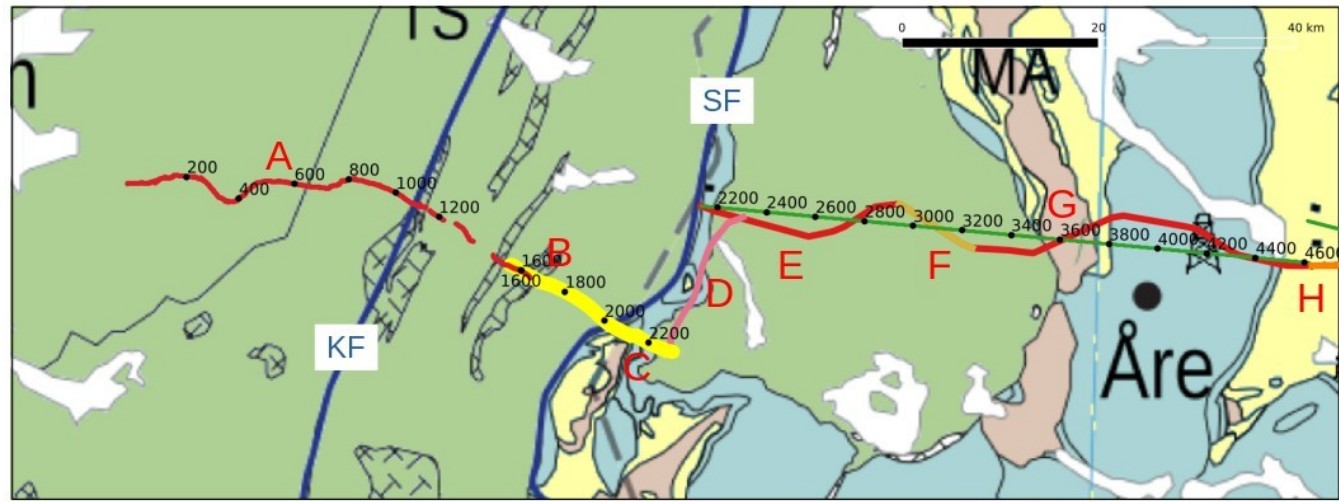

**Figure 2: Detailed geometry showing the overlap of segments A and B on the Norwegian side and the connection between segment C to segment E via segment D on the Swedish side. Note that the coordinates plotted for segments A, B and C are midpoints as calculated from source and receiver midpoints in the SEGY headers. CDP coordinates were not available in the headers, therefore we assumed that CDP bins were populated by traces whose midpoints were located to the nearest CDP. This implies that the**
120 **plotted CDP locations may differ somewhat from the ones actually used. Geological map based on Lescoutre et al. (2022a).**

**4 Method**

It would have been advantageous to have merged the source gathers from all the segments into one data set and reprocessed the combined parts as a single profile. However, segments A, B and C were only available as stacked sections in SEGY file format, while segment D was only available on paper in the report by Palm et al. (1991). Segments E to H were previously
processed as a single profile and presented in Juhojuntti et al. (2001), as were segments I and J. Due to these constraints, the previously processed data have been used in this contribution, except for segment D which was digitized and reformatted to SEGY as described in Sopher (2018). Note that segments E to J are still available as raw source gathers. Given that the maximum receiver spacing is 50 m (resulting in a CDP spacing of 25 m) we have resampled segments A, B and C to a trace spacing of 25 m on the unmigrated stacked sections. The resampling to 25 m for these segments was done by converting the
SEGY files to grids (segment A had a trace spacing of 20 m and segments B and C trace spacings of 12.5 m) and then resampling using GMT (Wessel and Smith, 1998). Segments E to J were already processed with a CDP spacing of 25 m. This provides a consistent trace spacing of 25 m on all the profiles that are included in the combined unmigrated stacked section which can be migrated as a single section.

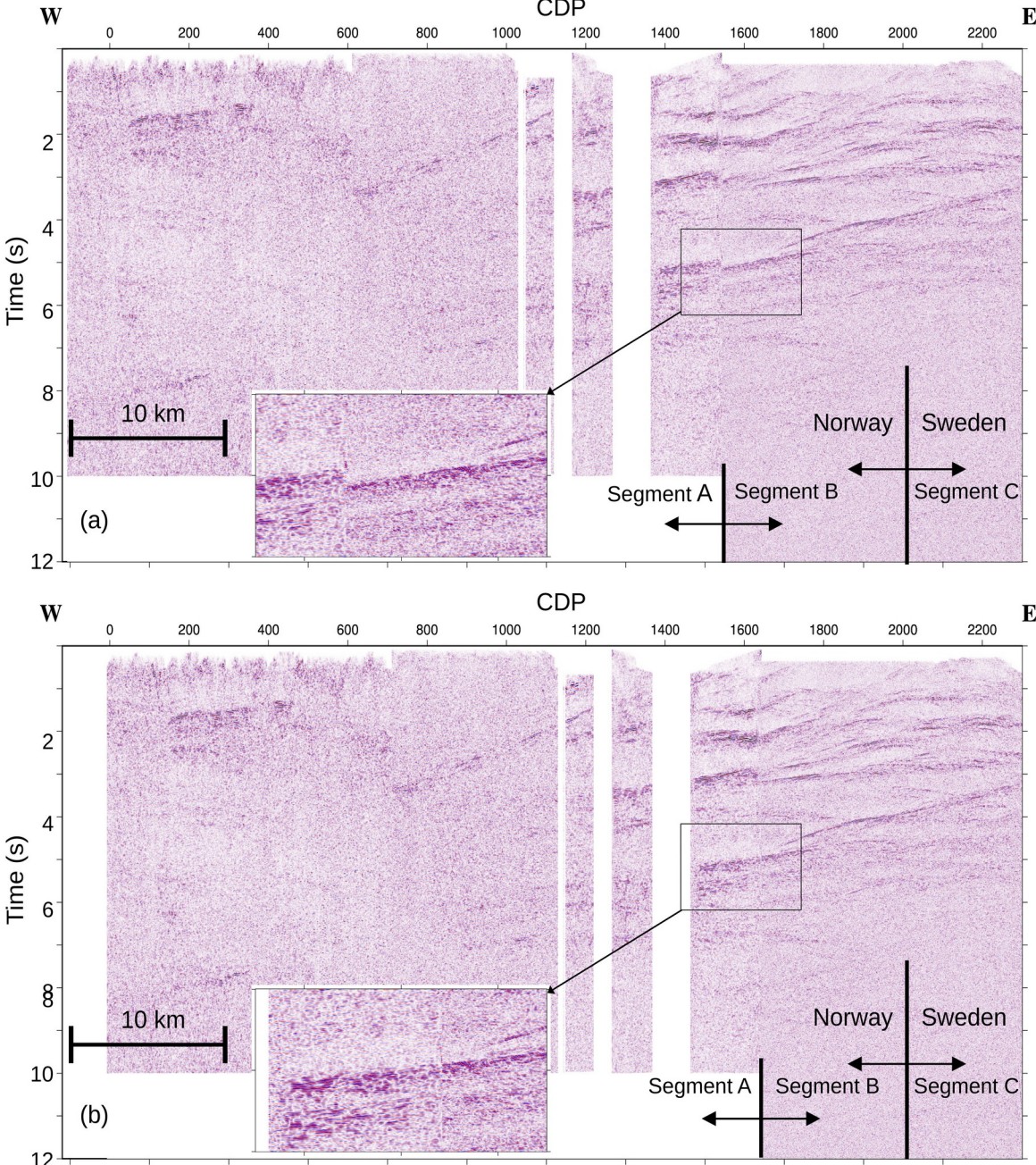

**Figure 3: (a) Merged segments A, B and C as done by Lescoutre et al. (2022a) and Roberts and Hurich (2018). (b) Merged segments A, B and C in this paper. The new section now shows more continuity between segments A and B. Rectangular boxes highlight an area where the new merging has significantly improved the image. Gaps in segment A are due to lack of subsurface coverage along those parts of the profile (See Figure 2).**

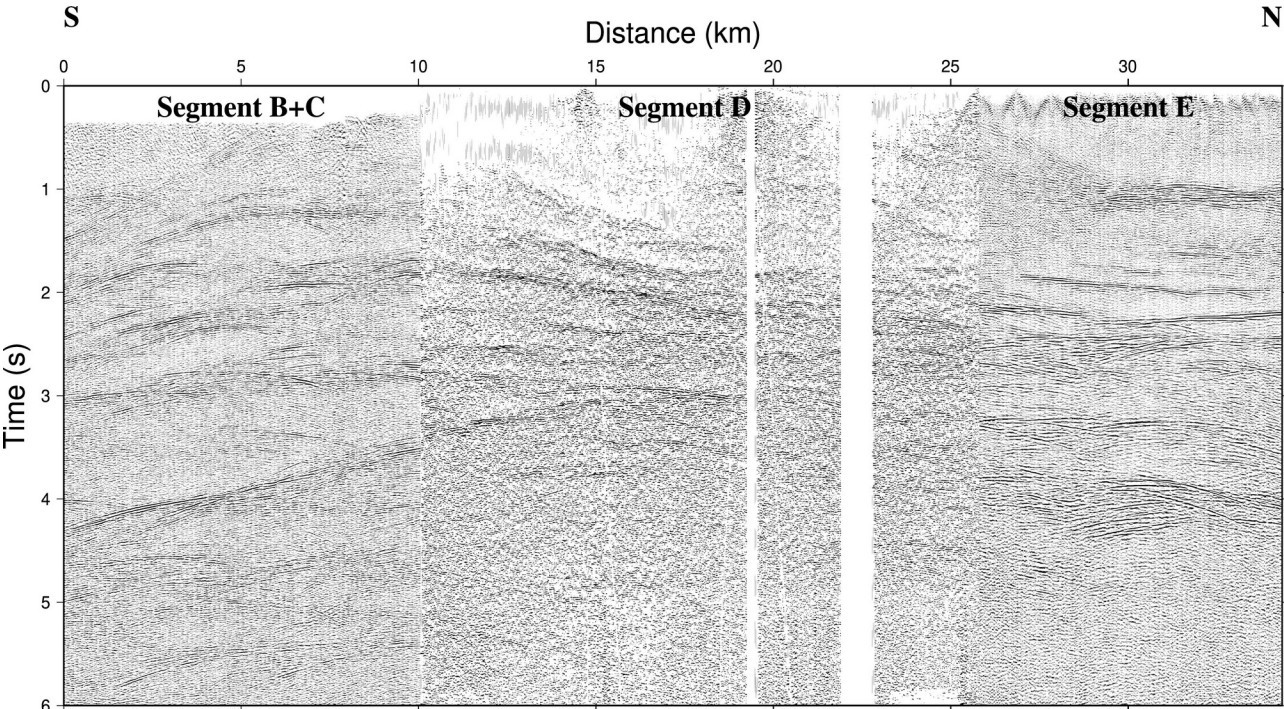

**Figure 4: Merged segments B, C, D and E. Segment D was only available on paper and was digitized from the report in Palm et al. (1991). Figure shows that there is an out-of-the-plane component to the data where segments C and D intersect. Therefore, it is not clear what is the best way to merge segment C with segment E to generate a continuous W-E directed profile to migrate. Note also the clear diffracted energy on segments B, C and E.**

As mentioned previously, the main differences between the present processing and previous versions are in the western part of the merged profile. Figure 3a shows how Lescoutre et al. (2022a) merged the Norwegian part of the CCT. This merging is similar to that presented in Roberts and Hurich (2018), but does not take into account the overlap between segments A and B (Figure 2). There is a clear break in the reflections where the two segments join. Based on Figure 2 there is about a 2.5 km overlap between segments A and B. Given that the data signal to noise ratio on segment A (closer source spacing, higher fold) is higher than segment B, we chose to retain the data from segment A and merge the two profiles about 2.5 km east of the western end of segment B (Figure 3b). The reflections are now nearly continuous across the merger, allowing better correlation between the two segments. Getting the merger point right becomes even more important for migration of the data since abruptly ending reflections on stacked sections will generate "smiles" on the migrated sections.

The most problematic area for providing a continuous seismic profile is how to merge segments C and E since there is a northward jump from the eastern end of C to the western end of E by about 15 km. Segment D connects these two profiles and provides some guidance (Figure 4). The reflections in the upper 2.5 s have a clear northward dip component on the western half of segment D and then flatten. The reflections between 2.5 s and 3 s are mainly sub-horizontal and the distinct one below 3 s has a southward dip on the western half, but then also becomes nearly horizontal towards the north. It is not possible to project segment C onto segment E in a consistent manner so that all reflections become continuous across the

merger point. We choose to merge the two segments about 4 km east of the western end of segment E since this provides the most continuous appearance (Figure 5), but noting that the upper 2.5 s should be interpreted with caution. Note the numerous

diffractions both on the eastern end of segment C and the western end segment E (Figure 4), indicating significant faulting in this area. These diffractions do not appear on segment D, probably because the faults have a roughly N-S strike. Finally, we migrate the data shown in Figure 5 using Stolt migration with a 1D velocity function starting at 5500 m/s at the surface and increasing to a RMS velocity of 6000 m/s at 3 s  and depth convert it with the same velocity function (Figure 6). Figure 7 shows a zoomed view of how diffractions collapse upon migration.

Note that the image of segment D in Figure 4 gives the impression that the conversion of the paper section to SEGY format resulted in a good quality digital section. This appearance is, however, deceiving. Attempts to migrate the section resulted in rather poor images with significant smearing and loss of detail. Therefore, we provide only the unimigrated version of segment D in this paper. If a larger format plot, such as A1 or A0, of segment D had been available then some post-stack processing of the section could perhaps have been performed after digitization, allowing a more detailed interpretation of the

structure at the border. Sopher (2016) shows an example where such post-stack processing of former paper sections has been successful on data from southern Sweden.

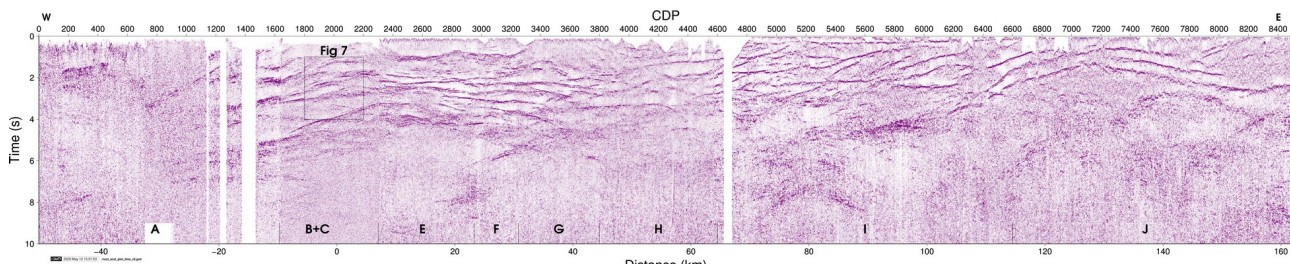

**Figure 5: Single stacked section of merged segments A to C and E to J as described in the text. CDP numbering corresponds to that shown in Figure 2 and letters corespond to the segments in Table 2. Section has been coherency enhanced prior to plotting.**

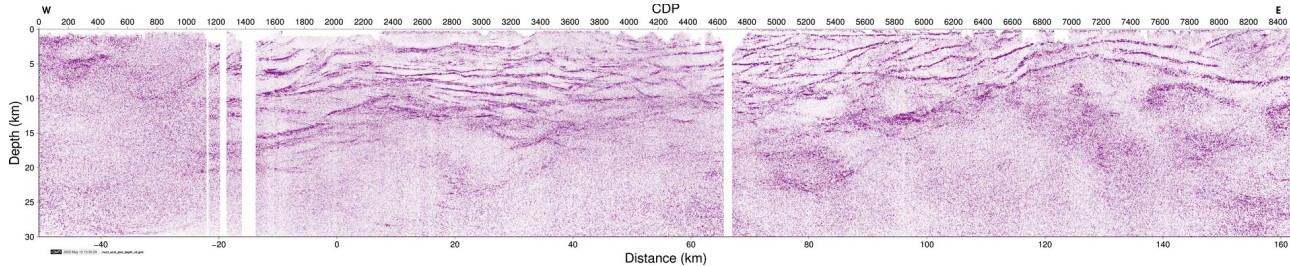

**Figure 6: Migrated and depth converted section of the stacked data in Figure 5 using a 1D velocity function starting at 5500 m/s at the surface and increasing to 6000 m/s at 3 s. CDP numbering corresponds to that shown in Figure 2. Section has been coherency enhanced prior to plotting.**

## 5 Results and discussion

The depth converted image (Figure 6) obtained west of the border has some significant differences compared to that of
Lescoutre et al. (2022a) and Roberts and Hurich (2018). Therefore, we focus our discussion on this part of the profile since
the interpretation of Lescoutre et al. (2022a) east of CDP 3000 remains unchanged. In the west the main differences are due
to migration in the CDP interval 1400 to 2200 where the diffractions have better collapsed, revealing shorter reflective
segments that appear to offset to one another (Figure 8d). Since most of these segments are either sub-horizontal or gently
west dipping this appearance is not due to the merging process.

Our new geological interpretation (Figures 8a and 8b) suggests that most of the flat-lying to moderately dipping reflections
can be interpreted as representing dolerites within the Precambrian basement, such as described in the central and eastern
part of the profile (Lescoutre et al., 2022a) and as observed in basement windows in the region (Johansson, 1980). Hauser
(1990) suggested early on that much of the reflectivity could be due to the presence of dolerite sills. In detail, some of these
reflections appear offset (top-to-the-east) along gently W-dipping reflections which, in contrast to the interpreted dolerites,
appear as continuous (sometimes very discrete) reflections from the western part to the central part of the profile. Such
reflections are thus interpreted, in accordance with previous interpretations (e.g., Gee and Stephens, 2020 and references
therein) and as suggested by the overall tectonic model of the area, as due to contractional structures related to the
Caledonian orogeny. We also observe some sub-horizontal reflections that show a normal offset across steep W-dipping
horizons (sometimes inconspicuous, Figure 8d).

In addition to the Kopperå fault as identified by Roberts and Hurich (2018) (marked by KF in Figure 8), our results highlight
the Steinfjell normal fault (SF in Figure 8) bounding the Skardöra antiform to the east (Sjöström and Bergman, 1989). Along
the Steinfjell fault, our reprocessing clearly images a thick deformation zone in its footwall with reflections showing
downward drag or bending near the fault (grey lines on Figure 8b and 8c). This fault seems to flatten at ~10 km depth.
Below the Skardöra antiform (CDP 1900-2100), our results highlight a newly identified extensional fault zone which offsets
the interpreted dolerites and supposedly the Caledonian thrust faults (e.g., at ~14 km depth). Its upward propagation is
unclear as shallow seismic reflections do not appear to be significantly offset.

In the hanging-wall of the Steinfjell fault, the strong reflections (interpreted as dolerites) show an upward dragging against
the fault (Figure 8c) and highlight the downward displacement of the parautochthonous basement west of the Skardöra
antiform. Note that this interpretation significantly diminishes the thickness of the Upper Allochthon to the west of the
Skardöra antiform (resulting in a shallower top basement depth) and reduces the extensional displacement along the
Steinfjell fault to about 3 to 5 km. Our interpretation also implies that the Middle Allochthon unit is locally discontinuous or
very thin in comparison to the interpretation by Hurich (1996). Reduced thickness of the allochthons in this area compared to
previous interpretations is consistent with recent potential field data interpretations by Olesen et al. (2025).

To summarize, our reprocessing provides a clearer image of the bedrock architecture in the western part of the central
Scandinavian Caledonides. It shows that the dolerite sill swarm propagates further west than generally previously interpreted

and reveals new structural features that question the overall nappe stack geometry and the contractional/extensional deformation in the parautochthonous basement.

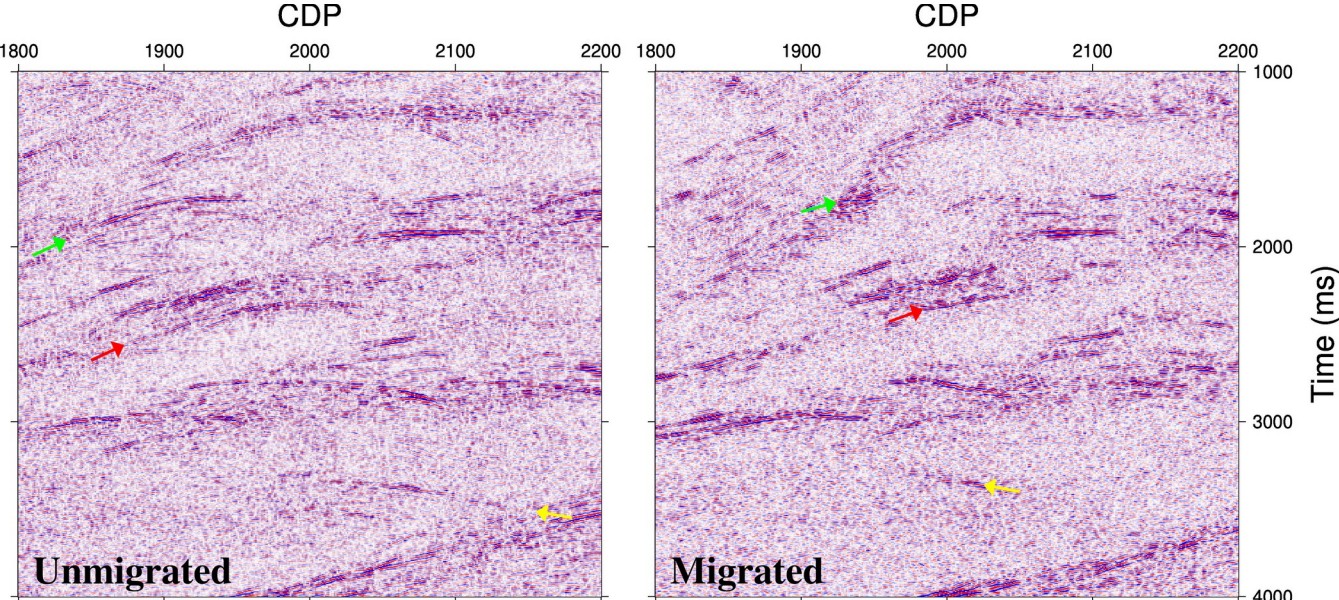

**Figure 7: Examples of diffractions that collapse after migration. Arrows are colour coded so that tails of diffractions in the unmigrated section migrate to a corresponding coloured "point" in the migrated section.**

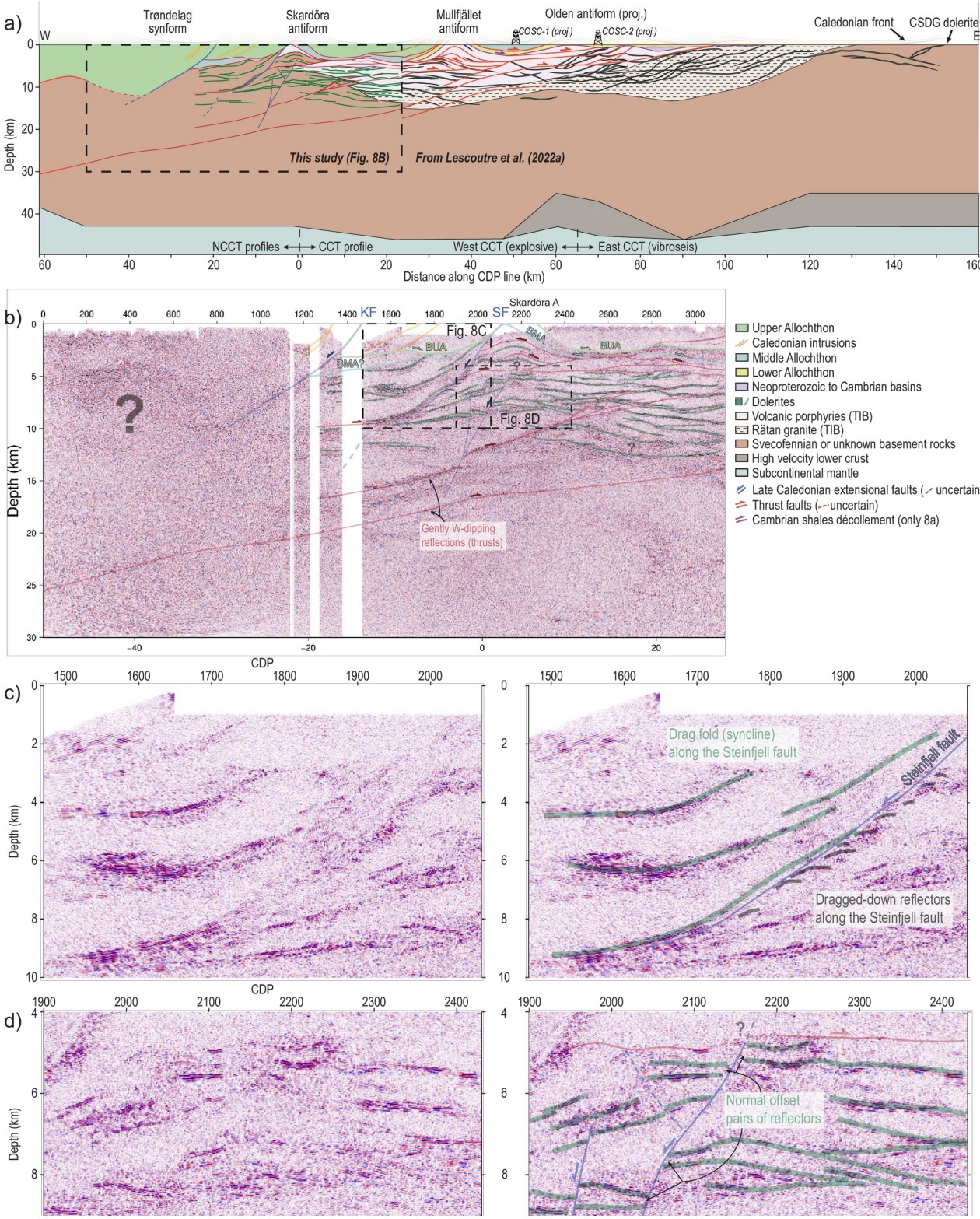

**Figure 8 (previous page): (a) Interpretation of the depth converted migrated merged profile. The section eastwards of distance 25 km is the same as in Lescoutre et al. (2022a) while the section west of this point is mainly new. COSC boreholes have been projected onto the figure. (b) The seismic section below shows in more detail how the interpretation was made. BUA – Base Upper Allochthon; BMA – Base Middle Allochthon; KF – Kopperå fault; SF – Steinfjell fault. (c) Zoomed view of the Steinfjell fault area. (d) Zoomed view of normal faulting in the Mullfjället antiform area.**

## 6 Conclusions

We present, for the first time, a complete migrated seismic section across the central Scandinavian Caledonides that includes available data both from the Norwegian side and the Swedish side. To produce the section different segments of the profile were resampled in space and time to a uniform interval. Gaps and overlap in the segments were carefully considered. Even though the original source gathers were not available for the Norwegian side and only a paper record from segment D existed we were able to produce a section which better represents the crustal structure across the Scandinavian Caledonides. Our work shows that vintage seismic data can still provide new insights when several campaigns are merged in a consistent manner. Although the paper record in this work was not of high enough quality to produce SEGY data that could be post-stack processed this may not be the case for areas where larger format paper sections are available. Our work shows that the preservation and retrieval of seismic data should be prioritized for future research.

Based on the depth converted migrated image a revised interpretation of the western part of the profile is made. In contrast to previous interpretations, we interpret the shorter sub-horizontal reflections in the Skardöra antiform to originate from faulted dolerite sills rather than shear zones. Results from the COSC-2 drilling further east support this interpretation where reflections of similar character are also present. It is likely that the dolerite system encountered there continues west into the Skardöra antiform. If correct, then our interpretation implies that the Upper Allochthon to the west of the Skardöra antiform is significantly thinner than suggested in previous studies and that the extensional displacement along the Steinfjell fault is about 3 to 5 km, less than previously inferred. Furthermore, the Middle Allochthon may be locally discontinuous or very thin west of the Skardöra antiform.

## Data availability

To request the data associated with this research, contact the corresponding author of the article after the publication of this work.

## Author contribution

CJ and RL conceptualized and designed this study. CJ was responsible for the data processing. RL, BA and CJ led the geological interpretation. CJ wrote the initial draft with input from RL and BA. All authors contributed to the results and discussion and approved the submission of this paper.

**Competing interests**

The authors acknowledge that there are no conflicts of interest.

**Acknowledgments**

We thank the Norwegian and Swedish Research Councils for their support in the 1980s and 1990s towards this work. We thank Chuck Hurich for making the seismic sections on the Norwegian side available for our use. We thank Daniel Sopher for converting the paper section of segment D to SEGY format. Globe Claritas™ under the academic license from Petrosys Ltd. and Seismic Unix was used for the data processing.

**Supplementary material**

Higher resolution versions of Figures 5 and 6 are presented in the supplementary material.

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
