# Peer review of "A new look at reflection seismic data from the Central Caledonian Transect across the Scandinavian Peninsula"

_EGUsphere, 2025_

## Author Response (AR1)

**RC1: 'Comment on egusphere-2025-1196', Andrew Calvert, 15 Apr 2025**

Deep seismic reflection profiling of the continental crust has been taking place for more than 50 years, and many of the early profiles are unlikely to be reacquired due to cost, changing environmental requirements and growing infrastructure. So it is important to preserve in a digital format as much of the early data as possible so that they can be easily displayed and subject to modern coherency enhancement techniques. In their short, well-organized paper, Juhlin et al. combine results from multiple short profiles across the Scandinavian Caledonides to create a new more readily interpreted section than previous results that will aid future workers in the area. My main suggestion is that greater support be provided using an additional blowup figure for the interpretation and the identification of downward drag of reflections near the Steinfjell fault, which I cannot see in the compressed Figure 7. If it is a significant improvement, it would also be helpful to include an additional coherency filtered version of Figure 6, which should be possible since the recovered data are now all in a digital format.

**Response: We have coherency filtered the data in Figures 5 and 6. We also provide zoomed views of the Steinfjell fault and the Skardöra anticline. The time and depth sections in the supplementary material are still the sections without coherency filtering.**

I have a few other additional suggested changes, questions and comments as indicated below:

40: Is the direction of subduction known?

**Response: We have added the following sentence: "Paleogeographic reconstructions (Torsvik and Cocks, 2017) suggest that subduction was mainly westward to weakly WNW at ~425-420 Ma and switching to W to weakly WSW at around 410-400 Ma"**

41: Citation needed for "returned and thrust onto Baltica"

**Response: We have added the following references: Arnbom (1980); Gee et al. (2008) and Majka (2014)**

47: What is meant by serpentiized gabbros? Surely you mean peridotites or metamorphosed gabbros?

**Response: We changed this sentence to "The Upper allochthon, including the Köli Nappe complex, consists of remnants of oceanic crust in the form of partly metamorphosed gabbros and metamorphosed sedimentary rocks  (i.e., phyllites)."**

Table 1: No lengths included for line segments  I and J

**Response: Lengths now included.**

84: Should there be a reference for CSDG?

**Response: CSDG is the abbreviation for Central Scandinavian Dolerite Group. We have now made this clear in the text.**

142: subhorizontal instead of "fairly horizontal"?

**Response: Changed to "mainly sub-horizontal"**

146: Change to "but noting that when interpreted the upper 2.5 s"

**Response: Changed to "but noting that the upper 2.5 s should be interpreted with caution."**

148: Change to " probably because the faults"

**Response: Changed**

149: I assume the migration and depth conversion velocity functions are the same here.

**Response: Yes, they are the same and we have made this clear in the text.**

Figure 5 caption: Single stacked section of merged segments A to C and E to J as described in the text

**Response: Changed**

Figure 6 caption: Migrated and depth converted section of the stacked data in Figure 5.

**Response: Changed**

165: (top- to-the-east) doesn't necessarily define thrusting. Best to be explicit about reverse motion if that is what you mean.

**Response: We don't understand this comment. The sentence l.165 describes the offset and geometry of the reflection along which the offset is observed (observation). In the next sentence, we explain that altogether (top-to-the-east offset across a W-dipping reflection) this supports the interpretation of a thrust fault, most probably related to the Caledonian orogeny.**

165: Label W-dipping reflection on figure so they can be clearly identified in the text.

**Response: Done**

173, 176: I was unable to locate the downward drag or bending near the fault (see earlier comment).

**Response: We now provide a zoom view of this area in Figure 8 (previously Figure 7) showing these features.**

Figure 7: Include legend for K, SF, BMA, BUA and any abbreviations used.

**Response: These abbreviations are defined in the caption. We feel that this is sufficient.**

227: insert line spacing before Gee et al reference

**Response: Done**

261: insert line spacing before Lehnert et al reference

**Response: Done**

271-278: There are two Lescoutre et al (2022) references. I assume they should be noted as 2022a and 2022b in reference list and text.

**Response: Done**

**RC2: 'Comment on egusphere-2025-1196', Don White, 05 May 2025**

General comments

This paper presents the results of a composite seismic profile across the central Scandinavian Caledonides. The original seismic data used in the study were acquired as much as 30 years ago and have been the basis for previous seismic interpretations in this region (e.g., Palm et al., 1991; Hurich 1996; Juhojuntti,et al. 2001; Lescoutre et al.2022; ). The ongoing utility of these data demonstrates the long-term value of archived seismic data. Previous interpretations of these data were based on the individual line segments or juxtaposition of separately processed seismic sections. The work presented here attempts to improve the merging of the profiles by uniformly resampling the individual data sets, and then suturing prior to migration in a 'best-fit' sense accounting for profile overlap and gaps. The resultant composite stack section is then migrated to provide a transect with more consistent imaging. Whereas the resultant seismic image is similar to previous results (cf. Lescoutre et al., 2022) it significantly improves the image along a 50 km segment in the western part of the transect. The authors focus on this western region of the profile and present an alternative interpretation that extends the presence of pre-collisional dolerite sills further west than previously identified and uses kinematic indicators in the images to propose significant post-collisional extensional structures. The interpretation is supported by the new data results and is generally consistent with other previous work done in this area while incrementally introducing/extending the knowledge of crustal architecture in the area.

In regard to improvements to the paper, I have included some suggestions in the specific comments below. For the most part, they consist on points of clarification and better identifying features in the figures that are described in the text. The manuscript is well-written and illustrated.

Specific comments

Method Section:

1. Add description of how the trace interpolation was done. Also, although the data are referred to as 'stacked' sections, it would be good to explicitly state that they are 'unmigrated'.

**Response: We added the following "The resampling to 25 m for these segments was done by converting the SEGY files to grids (segment A had a trace spacing of 20 m and segments B and C trace spacings of 12.5 m) and then resampling using GMT (Wessel and Smith, 1998)." and now explicit state they are unmigrated.**

2. An alternative to the 2D approach taken in the vicinity of the segment C-D-E line jog would be to explicitly consider the 3D geometry. For example, sections A-C, D, and E-J could have been migrated separately, and then 3 segments could be merged in a fence diagram to show the 3D structure. I recognize there would be limitations on the quality of the migrations for shorter line segments (especially for line D), but was such an approach considered or do you think it would be viable?

**Response: The purpose of Figure 4 is to show the continuity of some of the reflections at the Norway-Sweden border and that there is some 3D structure. Migration of the individual segments would result in reflections not tying at the intersections of the segments, leading to a misrepresentation of the continuity. Therefore, we prefer to present the unmigrated segments showing that most of the reflections tie at the intersections and there are out-of-the-plane components, especially on the Norwegian side.**

Results and discussion

1. On pg. 9, it is stated that one of the main differences obtained in the new result is "due to the collapse of diffractions" during migration. Why were diffractions not collapsed in any of the earlier analyses of Lescoutre et al. (2022) or Roberts and Hurich (2018). Clarify.

**Response: We agree that this sentence is misleading. The problem is that we have not been able to locate a published migrated version of segments A to C in high enough resolution to compare with. Roberts and Hurich (2018) indicates that the diffractions have not completely collapsed. Lescoutre et al. (2022) never migrated segments A to C. We now write that the "diffractions have better collapsed".**

2. In addition to Fig. 7, some of the key seismic-based interpretation features should be expanded in separate figures to allow the reader to assess the evidence for these important features of the interpretation. For example, I would suggest expanded panels showing:

-diffractions before and after collapse by migration, and 'have collapsed, revealing shorter reflective segments that appear offset to one another.'

**Response: We have added a new figure (Figure 7) showing how the diffractions collapse and result in short reflecting segments.**

-'sub-horizontal reflections that show a normal offset across steep W-dipping horizons (sometimes inconspicuous).'

**Response: We have added a zoom in Figure 8d (previously Figure 7) showing these offsets.**

-the 'Steinfjell fault, ... a thick deformation zone in its footwall with reflections showing downward drag or bending near the fault'

**Response: We have added a zoom in Figure 8d (previously Figure 7) showing these features.**

-'In the hanging-wall of the Steinfjell fault, the strong reflections (interpreted as dolerites) show an upward dragging against the fault'

**Response: We have added a zoom in Figure 8d (previously Figure 7) showing these features.**

Figures

Fig. 1 or 2: overlap between segments A and B is not obvious. Either label or highlight with line thickness.

**Response: We have increased the thickness of segment B so it should now be clear where the overlap is with A.**

Fig. 1: add COSC drill location symbols (derricks) to the legend.

**Response: Done**

Fig. 2 caption needs clarification regarding midpoints vs. CDP

**Response: We added these sentences to the caption "CDP coordinates were not available in the headers, therefore we assumed that CDP bins were populated by traces whose midpoints were located to the nearest CDP. This implies that the plotted CDP locations may differ somewhat from the ones actually used."**

Fig. 3: comparing with Fig. 1/2 where the line geometry is shown, it is not clear what the 3 gaps in the seismic sections for segment 'A' are due to? There only appear to be 2 locations in Fig. 2 where there may be gaps in the seismic line. Clarify what the gaps in the seismic sections are due to.

**Response: Gaps in segment A are due to lack of subsurface coverage along those parts of the profile. The short 3$^{rd}$ gap in the section is smeared out due to the slight differences between midpoint locations and CDP locations.**

Fig. 4 Would be helpful to the reader to add some labels identifying features referred to in the caption (e.g., 'diffractions') or in the text.

**Response: We have added a new figure (Figure 7) showing examples of diffractions in the unmigrated section and how they collapse in the migrated section.**

Fig. 7 Legend shows the 'Cambrian shales decollement' as a purple arrow; I don't see this on the figure.

**Response: The purple arrow is present in (a) at the base of the Lower Allochthon. We have modified the caption to make it clearer.**

Technical corrections

Line 185: remove hyphen from parautochthonous.

**Response: Done**

Fig. 4 caption: 'that is not clear' should be 'that it is not clear'

**Response: Done**

**RC3: 'Comment on egusphere-2025-1196', Sofie Gradmann, 05 May 2025**

Juhlin et al. undertake a significant effort to reprocess a segmented key seismic reflection line from the central Caledonides across Norway and Sweden. These vintage data are of variable quality and some segments were initially only available as paper version or stacked data. The only possible joint processing step is a post-stack depth migration, which has been performed after re-gridding of the datasets to a common CPD-spacing. The final depth-converted seismic section yields more continuous reflections, which are re-interpreted using new insights from recent nearby drilling and reflection seismic campaigns.

The paper gives a very nice and thorough overview over the entire Central Caledonian profile. It is very helpful to see the success and limitations of the attempt to jointly process and interpret the segments. Much can still be learned from these old data sets through the eyes of modern processing techniques and newer geological insights – although it becomes clear that some of the old data is simply of too poor quality.

The manuscript is very well written and scientifically sound. I think it would still benefit significantly from (1) a better comparison of the new processing result with previous results, (2) a more thorough separation of interpretations (thrust, dolerite sills) and results (more and less continuous reflections) and (3) a better presentation of certain information in the figures.

(1) Comparison of new and previous results

The reader can barely see how the new merging of stacked data (Figure 4) improved the continuity of the reflectors. Can you add 1 or 2 close ups from the improved parts?

**Response: We assume it is Figure 3 you are referring to. We have now added zooms to show that our merging has improved the continuity of the reflectivity.**

The reader cannot see how the migration improved the unmigrated section without studying it on his/her own with the help of the supplementary material. Can you add a close-up of a key part (perhaps the same as in Figure 7b but uninterpreted for both stacked and migrated data)?

**Response: We now show zooms in Figure 8 (previously Figure 7) for the migrated and depth converted data. We do not see the point in interpreting the unmigrated data in this paper.**

If I understand correctly, the current paper applies a post-stack Stolt time migration to the entire section. Juhojuntti et al. (2001) apply a pre-stack Kirchhoff time migration to the Swedish section (segments E-J). Why was the Stolt migration chosen and how do the results differ from the pre-stack migrated Swedish section? A close-up of a representative subsection would be helpful.
(2) Separation of results and interpretations

**Response: Juhojunnti et al (2001) applied post-stack migration in their work. We use the same stacked section (unmigrated) as input into our study. The only difference is that we now apply post-stack migration to the entire profile, including the data on the Norwegian side presented in this paper.**

I generally agree with the interpretations brought forward. But these are often difficult for a reader to follow who is less familiar with the region.

The interpretation presented for final seismic section in Figure 7 is highly complex. It is not clear which parts are derived from new insights gained in this study and which are taken from other

studies. In the text, however, the authors clearly describe which new interpretations they derive from the new section. I suggest linking the reflections and characteristics described in ll 156-168 and onward to a separate Figure showing the non-interpreted seismic section with e.g. arrows pointing to the described features. The following figure (current Fig 7) then adds the interpretation of the reflections (dolerite, normal fault, thrust fault) etc.

**Response: We have now added two zoom views of the interpreted section to Figure 8 (previously Figure 7) showing more detail our interpretation in key areas in the western part of the section. We have also added the information (Figure 8) to better clarify which part of the seismic section interpretation is new (this study) and what is from Lescoutre et al. (2022a).**

Some interpretations are clearly not based on the seismic data alone but on other publications and tectonic models. An example are the kinematic indicators added to some reflections (interpreted as thrusts), in particular below the Skardöra anticline. Also, the downward drag or bending near the Steinfjell fault could be disputed, as well as the very deep thrusts and their continuation. I think a few more question marks or dashed lines would be appropriate here, together with a sentence in the discussion that the seismic data at hand simply does resolve certain features.

**Response: We have added a sentence and references in the discussion, pointing out that the interpretation as thrust faults is mostly based on previous interpretations and supported by the well-known nappe stack (eastward-directed) tectonic model. Regarding the dashed lines we have added a few where we were more uncertain. However, we think that this is part of the seismic interpretation to propose a structural/geological model that better explains the reflections and the surface geology, although this is probably not a unique solution. We already displayed questions marks where we are uncertain in critical parts of the interpretation (e.g., upward propagation/termination of the normal faults below the Skardöra anticline). We have added panels (Fig. 8C and D) with zooms to highlight some key reflection geometries and our interpretation.**

Several interpretations are adopted from Lescoutres et al. It needs to be clear in the figure which ones go beyond the earlier interpretations.

**Response: It is in the western part of the section, west of CDP 3000 that the interpretation differs from Lescoutre et al. (2022a). We have added a sentence to make this clear "Therefore, we focus our discussion on this part of the profile since the interpretation of Lescoutre et al. (2022a) east of CDP 3000 remains unchanged." We have also clarified this point on the new Figure 8A.**

(3) Improvement of figures

The choice of colours for the units in figures 1 & 2 is somewhat unfortunate (difficult to differentiate between the allochthons and between the basement units). If it's not too much work, I'd suggest using more colours.
Add 'Norway' to the overview figure
Enlarge legend (colors/patterns, line thickness for faults and font size)
Indicate faults/contact better. All contacts are currently erroneously indicated as 'Thrust faults'.
Label Kopperå and Steinfjellet faults.

**Response: Done**

Remove segment names and CDP numbers from Figure 1. Rather use the common transect names NCCT, CCT-west and CCT-east here

**Response: We have removed the CDP numbering from Figure 1, but choose to retain the segment names since we refer to these in the text and the labeling is important for understanding how the segments were merged.**

Explain abbreviations in Figure caption 1 (TS, SA, PF, MA, COSC drill sites). Move description of seismic segments to Figure 2.

**Response: We have added TS, SA, PF, MA to the figure caption and the drill sites to the legend.**

Add at least a black frame to Figure 2. It looks a bit lost without coordinates or frame

**Response: Done**

Indicate segments on Figure 2. Currently it is not clear where A or B end and C starts. G and H also have the same color.

**Response: Done**

Enlarge CDP numbers

**Response: Done**

Figure 3 – why not present this part with the new, corrected CDP numbers?

**Response: We have now renumbered to the new CDP numbers.**

Figure 3 – what do the arrows across the segment boundaries indicate? Area of overlap?

**Response: The arrows indicate which segments are which in the merged section.**

Figures 3-6 – add or enlarge horizontal scale (in kilometers)
Figure 6 – indicate extent of line segments A-J

**Response: Done**

Figure 5-6: Why is there a gap between segment H and I (CDP 4700)? Lescoutre (2022) do not show this gap. One would on the other hand expect a gap between segments C & E, where 'D' is left out.

**Response: Lescoutre et al. (2022) did not take into account the gap when merging the segments. See Juhojuntti et al. (2001) where the gaps is present. There is no gap between C and E since we are projecting C onto E.**

Figure 7 – Caledonian intrusions and dolerites are marked as lines not polygons, please adjust the legend.

**Response: Done**

In order to highlight the contribution of this paper, you could expand the legend with 'faults (this study)' or 'dolerites (this study)' and give them a different color or dashes. See also comment (2)
(4) Other comments

**Response: Having a different color/appearance would make the figure even more complex. We indicate in the figure what is from Lescoutre et al. (2022a) and what is from this study. Basically, all the interpretation west of CDP 3000 has been reworked, following the interpretation of Lescoutre et al. (2022a) to the east.**

L65: The Alum shales are no longer present below the nappes on the Norwegian part of the profile. So the sentence might need to be slightly modified to something with "… marks the sole thrust of the nappe stack in the west."

**Response: Changed "… marks the sole thrust of the nappe stack in the east."**

L70: Reference to Saalmann 2021 is missing

**Response: Reference was in the list, but without a line separating it from the one above.**

L74: "made up of mainly of granitoids" --> "consists of mainly granitoids"

**Response: Changed**

L85: Add reference for Mullfjället Antiform

**Response: Robinson et al. (2014) added.**

L128: Explain what features indicate a 3D structure

**Response: We changed this wording to "Figure shows that there is an out-of-the-plane component to the data where segments C and D intersect." since we do not know for sure that the structure is 3D, only that the profiles are not dip lines.**

L181: Refer to Olesen et al (2025) who also derive a relatively little depth to basement from potential field data in this area. https://doi.org/10.1144/SP557-2024-113

**Response: We added a sentence "Reduced thickness of the allochthons in this area compared to previous interpretations is consistent with recent potential field data interpretations Olesen et al. (2025)."**

L201 Skärdöra --> Skardöra

**Response: Changed**

**EC1: 'Comment on egusphere-2025-1196', David Snyder**

Three reviewers and I have now read your manuscript and agree that it is fundamentally sound and generally well written. The demonstration of the value of legacy data was especially noted. Each reviewer does make substantive suggestions that will much improve the clarity of the text and figures. I would like to particularly encourage you to make more use of and reference to the drill hole, ranging from locating it in every relevant map or section to showing any 'lithological core' or geophysical logs available. Given the novelty of the paper record recovery that you describe here, I would also very much want to have included more emphasis and elaboration on what new was learned from its reprocessing, lessons learned during this reprocessing and reinterpretation, and some recommendations as to when and where it would work elsewhere in the world. Include these in the Conclusions.

EC1 Comment: "I would like to particularly encourage you to make more use of and reference to the drill hole, ranging from locating it in every relevant map or section to showing any 'lithological core' or geophysical logs available."

**Response: Results from the COSC boreholes relevant to seismic interpretation have already been published or are in the process of being written up. Furthermore, the merged seismic profile presented in this paper does not pass over these boreholes. Therefore, we do not feel that we should include data from these boreholes in this publication. We have added the following sentences in the Introduction to make the reader aware of this. "Our new interpretation is aided by incorporating results from the two c. 2.3-2.5 km deep COSC boreholes that were drilled in the Swedish Caledonides in recent years (Lorenz et al., 2015; Lorenz et al., 2022). Even though the profile presented here does not directly pass over the boreholes we can make use of important observations from these boreholes. In particular, the strong reflections from the Precmabrian basement observed on a high resolution seismic profile (Juhlin et al., 2016) passing over the COSC-2 borehole are generated by dolerites that have intruded into highly homogenous volcanic rocks (Lorenz et al., 2022; Lescoutre et al., 2022a)." We also now show the projection of the borehole locations onto Figure 8a, showing that they are not so relevant for the revised interpretation in the west.**

EC1 Comment: "Given the novelty of the paper record recovery that you describe here, I would also very much want to have included more emphasis and elaboration on what new was learned from its reprocessing, lessons learned during this reprocessing and reinterpretation, and some recommendations as to when and where it would work elsewhere in the world. Include these in the Conclusions."

**Response: Recovery of paper records has been possible for quite some time, for example tif2segy in the Seismic Unix package. We used a more advanced routine provided by Sopher (2016). The image looks good in the paper, but is not of high enough quality for post-stack processing. We have added some sentences in the Method section and Conclusions section recording our experience in working with the vintage seismic data. In the Method section we write "Note that the image of segment D in Figure 4 gives the impression that the conversion of the paper section to SEGY format resulted in a good quality digital section. This appearance is, however, deceiving. Attempts to migrate the section resulted in rather poor images with significant smearing and loss of detail. Therefore, we provide only the unimigrated version of segment D in this paper. If a larger format plot, such as A1 or A0, of segment D had been available then some post-stack processing of the section could perhaps have been performed after digitization, allowing a more detailed interpretation of the structure at the border. Sopher (2016) shows an example where such post-stack processing of former paper sections has been successful on data from southern Sweden." In the Conclusions**

section we write "Even though the original source gathers were not available for the Norwegian side and only a paper record from segment D existed we were able to produce a section which better represents the crustal structure across the Scandinavian Caledonides. Our work shows that vintage seismic data can still provide new insights when several campaigns are merged in a consistent manner. Although the paper record in this work was not of high enough quality to produce SEGY data that could be post-stack processed this may not be the case from areas where larger format paper sections are available. Our work shows that the preservation and retrieval of seismic data should be prioritized for future research."